# Home-Based High-Intensity Interval Exercise Improves the Postprandial Glucose Response in Young Adults with Postprandial Hyperglycemia

**DOI:** 10.3390/ijerph19074227

**Published:** 2022-04-01

**Authors:** Yuto Nakayama, Kumiko Ono, Junya Okagawa, Junji Urabe, Ryoga Yamau, Akira Ishikawa

**Affiliations:** 1Graduate School of Health Sciences, Kobe University, 7-10-2, Tomogaoka, Suma, Kobe 654-0142, Hyogo, Japan; jun.aqua75@gmail.com (J.O.); baseex0719@gmail.com (J.U.); ishikawa@bear.kobe-u.ac.jp (A.I.); 2School of Medicine Faculty of Health Sciences, Kobe University, 7-10-2, Tomogaoka, Suma, Kobe 654-0142, Hyogo, Japan; 1843914m@stu.kobe-u.ac.jp

**Keywords:** high-intensity interval exercise, postprandial hyperglycemia, nausea, exercise

## Abstract

Postprandial hyperglycemia can be corrected by exercise; however, the effect of home-based high-intensity interval exercise (HIIE), a new time-efficient exercise, on glycemic control is unclear. This study aimed to investigate the effect of home-based HIIE on postprandial hyperglycemia. Twelve young adult males (mean age: 24.3 ± 2.3 y) with postprandial hyperglycemia that had not yet led to diabetes completed home-based HIIE, moderate-intensity continuous exercise (MICE), and control conditions on separate days, randomly. The intervention began 30 min after the start of a standardized meal intake, with 11 min of HIIE completed at maximal effort in the home-based HIIE condition, 30 min of running performed at 50% maximum oxygen uptake in the MICE condition, or 30 min of sitting at rest completed in the control condition. The participants sat at rest after each intervention for up to 120 min. Interstitial fluid glucose concentrations were measured using a continuous glucose monitoring system that scanned every 15 min for up to 2 h after the meal. The glucose concentrations after the meal were significantly lower in the home-based HIIE and MICE conditions than in the control condition (*p* < 0.001). There were no significant differences in the glucose concentrations between the home-based HIIE and MICE conditions. In conclusion, home-based HIIE was able to correct postprandial hyperglycemia.

## 1. Introduction

Postprandial glucose levels are more strongly associated with the risk of diabetes, cardiovascular events, and all-cause mortality than fasting glucose and glycosylated hemoglobin (HbA1c) levels in diabetic and non-diabetic individuals [1,2,3,4]. Even in healthy individuals, high postprandial blood glucose levels increase oxidative stress, cause vascular endothelial dysfunction, and increase the risk of developing cardiovascular disease [5,6]; thus, even non-diabetic individuals should pay attention to their postprandial blood glucose levels. Two-hour postprandial blood glucose levels <140 mg/dL are associated with a decreased incidence of metabolic syndrome and cardiovascular events [7,8].

Physical activity helps optimize blood glucose control [9]; for example, 30 min of moderate-intensity walking [10] and 10 min of stair climbing at a comfortable pace [11] have both been found to sufficiently reduce postprandial glucose levels. However, these exercises take a long time or require a specific location, and they do not solve the problem of “lack of time,” which is one of the factors that inhibit exercise [12]. In addition, the emergence of the severe acute respiratory syndrome–associated coronavirus 2 virus at the end of 2019 affected daily life worldwide, especially regarding physical activity, with reports of decreased physical activity in various countries compared with pre-coronavirus disease 2019 (COVID-19) pandemic levels [13,14,15]. Although the COVID-19 pandemic has been reported to have no effect on glycated hemoglobin in diabetic patients, there is a relationship between decreased physical activity and increased glycated hemoglobin caused by telecommuting [16]. In fact, it has been reported that the amount of physical activity in diabetic patients is decreasing [17], and it is not certain what the long-term effects will be. Considering the disincentive for physical activity in the past and the social situation in recent years, to continuously correct postprandial hyperglycemia, we propose that exercise should be time-efficient and able to be performed anywhere.

Therefore, this study focuses on high-intensity interval exercise (HIIE), which can be performed in a limited space at home (home-based HIIE (HBHIIE)). HIIE is a type of exercise that alternates between high intensity and low intensity for a short period of time, and various previous studies have compared it with moderate-intensity continuous exercise (MICE). According to recent research, HIIE is superior to MICE in terms of time efficiency, improvements in maximal oxygen uptake [18] and glucose metabolism [19], and enjoyment [20], and HIIE performed at home exhibits similar effects on cardiopulmonary function [21]. We developed an 11-min HBHIIE consisting of exercises that can be performed in a limited space, in consideration of the recommended home-based exercises under the COVID-19 pandemic [22]. In this study, we hypothesized that 11 min of HBHIIE would be as effective as 30 min of MICE in correcting postprandial hyperglycemia in young adults. The purpose of this study was to compare the effects of HBHIIE on postprandial hyperglycemia with those of MICE. If short HBHIIE can correct postprandial hyperglycemia as well as MICE, it may have a significant impact on the prescription of exercise, especially for individuals who feel reluctant to exercise or do not have time to exercise.

## 2. Methods

### 2.1. Participants and Screening Procedure

Thirty-seven healthy young adult males (mean age: 24.3 ± 2.3 y) participated in the screening session. In the screening session, participants were invited to the laboratory from 8:00 to 10:00 am following overnight fasting and performed self-monitoring of blood glucose, followed by consumption of a standardized meal containing 405 kcal (71 g, 70.5% complex carbohydrates; 4 g, 7.9% fat; 21 g, 20.9% protein; 1.5 g, 0.7% fiber). Twelve participants who had postprandial hyperglycemia, defined as a blood glucose level >140 mg/dL at 30 or 45 min after a meal, participated in the subsequent familiarization and experimental trials. This study was approved by the Kobe University Graduate School of Health Sciences Ethics Committee (approval number: 974-1), and written informed consent was obtained from all participants in accordance with the Declaration of Helsinki.

### 2.2. Familiarization Trials

First, the participants completed the International Physical Activity Questionnaire [23], and their physical measurements were recorded. Second, participants performed an incremental trial with a treadmill (GE Healthcare Japan Corp., Ltd., Tokyo, Japan) using the United States Air Force School of Aerospace Medicine protocol [24] to estimate maximal oxygen uptake (VO_2_max). On a different day, participants performed a second incremental trial on the treadmill to determine their individual running speed that corresponded to 50% VO_2_max. The participants started walking at an initial speed of 3 km/h. The treadmill speed was increased every 4 min by 1 km/h, up to a maximum of 8 km/h. Oxygen uptake (VO_2_) and heart rate (HR) were monitored using a gas analyzer (Aero Monitor AE-310S; Minato Medical Science Co., Ltd., Osaka, Japan) and a heart rate monitor (LRR-03; GMS Japan Co., Ltd., Tokyo, Japan), respectively. After the second incremental trial, participants participated in the HBHIIE familiarization trial, which involved watching the HBHIIE video clip and practicing the HBHIIE. Finally, a continuous glucose monitoring device (CGM) (FreeStyle Libre Pro; Abbott Japan Co., Ltd., Tokyo, Japan) was placed on their upper arms.

### 2.3. Experimental Trials

Figure 1 presents a summary of the experimental trials. A randomized crossover design was used in this study. Participants underwent three randomly ordered trial conditions at the same time on different days separated by a 48-h washout period. The three trial conditions were conducted within 2 weeks so that the passage of time would not affect blood glucose levels or body composition between trials. In all trials, participants sat and rested for 15 min first, after which they consumed the standardized meal containing 405 kcal (71 g, 70.5% complex carbohydrates; 4 g, 7.9% fat; 21 g, 20.9% protein; 1.5 g, 0.7% fiber) at a normal pace. After 30 min of sedentary rest following the start of food intake, participants underwent one of the three 30-min trial conditions described below. At the end of the trial, the participants took a 60-min sedentary rest.

### 2.4. Trial Conditions

The interventions administered 30 min after the meal consisted of three conditions: H, M, and C. In the H condition, participants performed HBHIIE for 11 min, followed by 19 min of sitting at rest. The HBHIIE consisted of fourteen 20-s bouts of high-intensity exercise (burpee jumps, squats, mountain climbers, high knees, jumping lunges, push-ups, and jumping jacks) separated by 20-s dynamic recovery phases. Each high-intensity phase was performed at maximal effort. In the M condition, participants performed moderate-intensity running at 50% VO_2_max for 30 min. In the C condition, the participants sat at rest for 30 min.

### 2.5. Measurements

Glucose levels were recorded automatically by the CGM every 15 min for 2 weeks, including during the three experimental trials. Glucose levels 2 h after eating the standardized meal on the trial days were compared among the three conditions. The trapezoidal rule was used to calculate the incremental area under the postprandial glucose curve (iAUC). Glucose concentration responses up to 2 h after lunch and dinner on the trial days were also compared among the three conditions.

The participants completed the Physical Activity Enjoyment Scale (PACES) [25] and Rating of Perceived Exertion (RPE) using the Borg scale [26] immediately after the HBHIIE and MICE programs. The visual analog scale (VAS) was used to assess nausea after each condition. The expired gas and HR were measured throughout each condition, and the total energy consumption was calculated using the obtained VO_2_ and carbon dioxide production values [27]. HR and VO_2_ were averaged every 10 s, and mean %HRmax and %VO_2_max were calculated as the average of these values expressed as a percentage of the maximum value. Peak %HRmax and %VO_2_max were calculated as the highest values of HR and VO_2_, respectively, averaged every 10 s as a percentage of the maximum value. VO_2_max represented the maximum value of VO_2_ estimated in the familiarization session, and HRmax was predicted by subtracting the age of the participants from 220.

### 2.6. Food Intake

The participants were asked to consume the same dinner on the day before each trial, lunch on the day of each trial, and dinner on the day of each trial and to record their meals and ingredient label on a form with a photograph. The carbohydrate content of the dinner on the day before each trial, lunch on the day of each trial, and dinner on the day of each trial was calculated based on the values on the ingredient label of the recording form. The participants were not allowed to consume alcohol 24 h before the intervention or engage in vigorous exercise while wearing the CGM for 2 weeks.

### 2.7. Statistical Analysis

All variables except RPE are presented as mean ± standard deviation, and RPE is presented as median ± standard error. All data were tested for normality using the Shapiro–Wilk test. One-way repeated measures analysis of variance (ANOVA) pairwise comparisons with Holm correction were used for the comparison of the iAUC and carbohydrate content of the three conditions. The differences in PACES, VAS, mean %HRmax, mean %VO_2_max, and total energy consumption for the H and M conditions were statistically analyzed using paired t-test, and differences in the RPE values were statistically analyzed using the Wilcoxon signed-rank test. The changes in glucose concentration from the start of food intake until the end of the 2 h trials were statistically analyzed by a two-way (time × condition) repeated measures ANOVA. Mauchly’s sphericity test was used to test for sphericity. If the assumption of sphericity was not met, Greenhouse–Geisser correction was applied to the number of degrees of freedom. Pairwise comparisons with Holm correction were performed if a significant difference was found. The partial eta squared (η_p_^2^) values were calculated to estimate the effect size. The level of statistical significance was set at *p* < 0.05. All data were analyzed using R 4.1.2 (The R Foundation, Vienna, Austria) for Windows (Microsoft, Redmond, WA, USA).

## 3. Results

### 3.1. Participant Characteristics

The participants’ characteristics are presented in Table 1. The mean fasting blood glucose level measured at the screening session was 93.4 ± 6.7 mg/dL, and the mean peak blood glucose level at 30 or 45 min after the standardized meal intake was 155.6 ± 6.7 mg/dL. The mean fasting blood glucose level of the participants was within the normal range (<100 mg/dL), and the mean maximum postprandial blood glucose level was greater than 140 mg/dL.

### 3.2. Exercise Intensities

The exercise intensities of the HBHIIE and MICE conditions are presented in Table 2. The total energy consumption was significantly higher in the MICE group than in the HBHIIE group (*p* < 0.001), and the mean %VO_2_max (*p* < 0.001), mean %HRmax (*p* < 0.001), and median RPE (*p* < 0.001) were significantly higher in the HBHIIE group than in the MICE group.

### 3.3. Food Intake

The carbohydrate content of the dinner on the day before each trial, lunch on the day of each trial, and dinner on the day of each trial is presented in Table 3. There was no significant difference in any of the carbohydrate contents.

### 3.4. Acute Glucose Response

Changes in glucose concentration up to 2 h after the standardized meal intake and the iAUC are shown in Figure 2. There was a significant interaction between time and condition for changes in glucose concentration (interaction: *p* < 0.001, η_p_^2^ = 0.5464; time: *p* < 0.001, η_p_^2^ = 0.7026; condition: *p* = 0.0054, η_p_^2^ = 0.3775). At 45 min (*p* = 0.0028), 60 min (*p* < 0.001), and 75 min (*p* < 0.001) after the start of food intake, glucose concentrations in the H and M conditions were significantly lower than those in the C condition. Conversely, at 90 min, only glucose concentrations in the H condition were significantly lower than those in the C condition (*p* = 0.0422). There were no significant differences between the H and M conditions at any time point. The iAUC was significantly lower in the H and M conditions than in the C condition (*p* < 0.001, η_p_^2^ = 0.6002), and there was no significant difference between the H and M conditions (*p* = 0.4261).

### 3.5. Chronic Glucose Response

Changes in glucose concentration up to 2 h after lunch and dinner on the day of each trial are shown in Figure 3. There was no significant interaction between time and condition for changes in glucose concentration up to 2 h after lunch (interaction: *p* = 0.0804, η_p_^2^ = 0.1251; time: *p* < 0.001, η_p_^2^ = 0.5832; condition: *p* = 0.1047, η_p_^2^ = 0.1855) and dinner (interaction: *p* = 0.4178, η_p_^2^ = 0.0825; time: *p* < 0.001, η_p_^2^ = 0.519; condition: *p* = 0.4245, η_p_^2^ = 0.0749).

### 3.6. Secondary Outcomes

The PACES and VAS scores for nausea are shown in Figure 4. The PACES values were not significantly different between the H and M conditions (*p* = 0.287). The VAS scores for nausea were significantly higher in the H condition than in the M and C conditions (*p* < 0.001, η_p_^2^ = 0.6303).

## 4. Discussion

In this study, we hypothesized that 11 min of HBHIIE would correct postprandial hyperglycemia as effectively as 30 min of MICE in young adults with postprandial hyperglycemia. As hypothesized, HBHIIE, with a duration of approximately one-third of MICE, corrected postprandial hyperglycemia. However, HBHIIE was more likely to induce nausea than MICE. In addition, the level of enjoyment of the exercise was not significantly different between the HBHIIE and MICE conditions.

To the best of our knowledge, this study is the first to report both the time efficiency and adequate postprandial glucose correction effect of HBHIIE. A previous study comparing the effects of 30 min of low-intensity exercise (LIE), moderate-intensity exercise (MIE), and HIIE on the acute glucose response in healthy males who perform inadequate physical activity found no significant difference between groups, even though the energy consumption was lower in the LIE group than in the MIE and HIIE groups [28]. In the present study, although the total energy consumption was significantly lower in the HBHIIE group than in the MICE group, there was no significant difference in the postprandial glycemic response.

It is not yet clear whether the effect of exercise on glucose regulation is simply a product of total energy expenditure. Time-efficient exercise may be beneficial for people with postprandial hyperglycemia in terms of exercise adherence if the total energy expenditure is low and the short-duration exercise is sufficient to regulate blood glucose levels. HBHIIE is more time-efficient than other exercises, such as 20-min walking [29], 24-min cycling HIIE [30], and 15-min resistance exercise [31], which have been reported to be effective in correcting blood glucose levels in a short period of time. In addition, a previous study that investigated 3-min stair climbing as a potential time-efficient exercise modality found that 3-min stair climbing cannot adequately correct blood glucose levels 30 min after a meal compared with 10-min stair climbing [11].

The 11-min HBHIIE performed after breakfast corrected the acute glucose response, but it did not affect the chronic glucose response after lunch and dinner. The 24-min cycling HIIE performed after breakfast in obese adults corrected the glucose response not only after breakfast but also after dinner [30]. High-intensity exercise has been shown to increase the secretion of growth hormone, which raises blood glucose levels, for 12.5 h afterward [32], and HBHIIE, a self-weighted resistance exercise-based maximal effort exercise, may have elicited a different hormonal response than previous studies due to differences in exercise style and intensity. In addition, in this study, we were unable to measure the amount of physical activity on the day of the experiment in each condition, and it is possible that the difference in the amount of physical activity between the conditions may have affected the glucose response.

It has been reported that HIIE is a more enjoyable exercise modality than MICE [33]; however, this study presented a different finding. The levels of enjoyment of the exercise were not significantly different between the HBHIIE and MICE conditions. As a potential explanation for these contrasting results, HIIE with high RPE has been associated with lower levels of enjoyment than HIIE with low RPE [34]. The HBHIIE program employed in this study was classified as a “very hard” exercise with an RPE of 17, indicating that it may have been too difficult to be enjoyable.

According to previous studies, exercises with frequent vertical movements are more likely to induce gastroesophageal reflux [35]. HBHIIE includes many jumping exercises, such as burpee jumps and jumping jacks, and these exercises are likely to induce nausea. In contrast, nausea induced by 60 min of high-intensity exercise immediately after a meal reached 25 on the VAS [36], whereas nausea induced by vection reached 43 on the VAS [37], suggesting that the VAS score of 25.1 observed for nausea induced by HBHIIE may not be high in comparison. However, no study has evaluated the effect of post-exercise nausea on exercise adherence; thus, further investigation is warranted.

HBHIIE was able to correct postprandial blood glucose levels in a short time, but the disadvantages of the exercise were clear: it was too strenuous to be enjoyable, and it tended to induce nausea. Exercise with high adherence is necessary to correct postprandial blood glucose levels that occur at every meal. To make HBHIIE more accessible, it is necessary to investigate HBHIIE without jumping events, with shorter exercise duration, and with lower exercise intensity. In addition, longitudinal studies are required to examine the adherence to this exercise type.

Our study has several limitations. To prevent the menstrual cycle from affecting blood glucose levels, only men were included in this study; however, studies including women are needed to generalize the results. We were not able to collect blood samples; hence, we could not examine the changes in hormonal responses that could affect blood glucose levels. In addition, interstitial fluid glucose concentrations tend to have a delayed response compared to blood glucose levels, and the CGM used in this study has shown that glucose concentrations tend to be low during exercise [38]. Therefore, the values of the interstitial fluid glucose concentration measured in this study may have been an underestimation of the actual glucose concentration compared with the measurement of blood glucose levels. For a more accurate assessment of the glycemic response, collecting venous blood samples is necessary in future studies.

## 5. Conclusions

HBHIIE was able to correct postprandial hyperglycemia even though the duration of exercise was about one-third of the MICE. In contrast, there was no significant difference in enjoyment between HBHIIE and MICE, and HBHIIE was more likely to induce nausea than MICE, but not to a high extent. Since exercise with high adherence is necessary to correct postprandial hyperglycemia, more enjoyable exercise regimes that are less likely to induce nausea require investigation.

## Figures and Tables

**Figure 1 ijerph-19-04227-f001:**
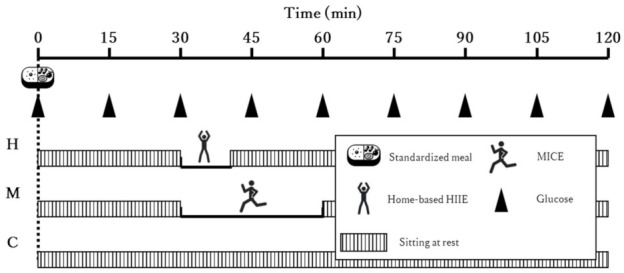
A summary of the experimental trials. Participants started eating the standardized meal at 0 min. After 30 min of sitting at rest, participants performed 11 min of home-based HIIE in the H condition, and 30 min of MICE in the M condition. The participants sat at rest until 120 min after the start of eating the standardized meal. In the C condition, the participants sat at rest for the entire 120 min. Glucose concentrations were automatically measured every 15 min.

**Figure 2 ijerph-19-04227-f002:**
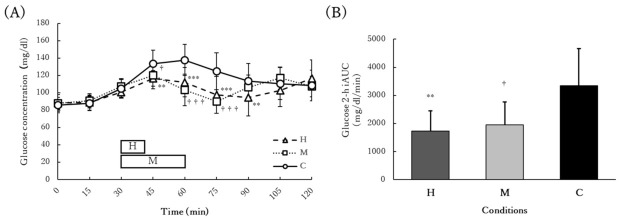
Changes in (**A**) glucose concentration up to 2 h after the standardized meal intake and (**B**) the iAUC. The participants performed 11 min of home-based HIIE 30 min after a meal in the H condition (H), 30 min of moderate-intensity exercise 30 min after a meal in the M condition (M), and 30 min of seated rest 30 min after a meal in the C condition (C). There were no significant differences between the H and M conditions. The H and M bars in panel (**A**) represent the timing of the exercises. ** *p* < 0.01 vs. C; *** *p* < 0.001 vs. C; ^†^ *p* < 0.05 vs. C; ^†††^
*p* < 0.001 vs. C. Abbreviations: iAUC, incremental area under the postprandial glucose curve.

**Figure 3 ijerph-19-04227-f003:**
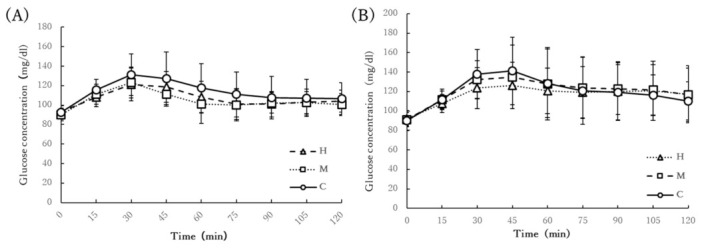
Changes in glucose concentration up to 2 h after (**A**) lunch and (**B**) dinner. The participants performed 11 min of home-based HIIE 30 min after a meal in the H condition (H), 30 min of moderate-intensity exercise 30 min after a meal in the M condition (M), and 30 min of seated rest 30 min after a meal in the C condition (C). There was no significant interaction between time and condition for changes in glucose concentration up to 2 h after lunch and dinner.

**Figure 4 ijerph-19-04227-f004:**
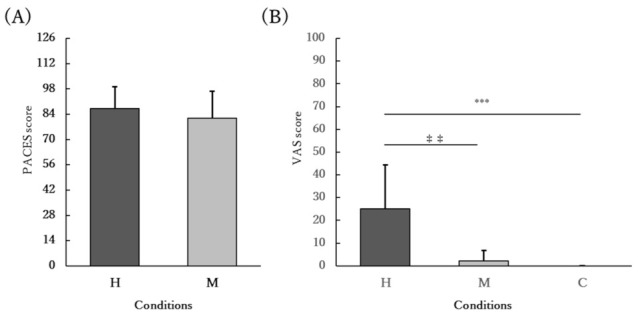
(**A**) PACES and (**B**) VAS for nausea. The participants performed 11 min of home-based HIIE 30 min after a meal in the H condition (H), 30 min of moderate-intensity exercise 30 min after a meal in the M condition (M), and 30 min of seated rest 30 min after a meal in the C condition (C). (**A**) The mean PACES value was not significantly different between the H and M conditions (*p* = 0.287). (**B**) The mean VAS score was significantly higher in the H condition than in the M and C conditions (*p* < 0.001). ^‡‡^ *p* < 0.01 vs. M; *** *p* < 0.001 vs. C. Abbreviations: PACES, Physical Activity Enjoyment Scale; VAS, visual analog scale.

**Table 1 ijerph-19-04227-t001:** Participant characteristics.

Variable	Values
Age (years)	24.3 ± 2.3
Height (cm)	173.3 ± 5.8
Mass (kg)	68.0 ± 10.4
BMI (kg/m^2^)	22.6 ± 2.3
Fasting glucose (mg/dL)	93.4 ± 6.7
Peak glucose (mg/dL)	155.6 ± 12.1
VO_2_max (mL/kg/min)	42.9 ± 3.8
MICE velocity (km/h)	5.8 ± 0.5

Data are presented as means ± SD. Fasting glucose and peak glucose were measured during the screening procedure, and peak glucose was the mean of the maximum glucose concentrations at 30 or 45 min after consumption of the standardized meal. MICE velocity represents the speed of the run performed in the M condition. Abbreviations: BMI, body mass index; VO_2_max, maximal oxygen uptake; MICE, moderate-intensity continuous exercise.

**Table 2 ijerph-19-04227-t002:** Exercise intensities.

Variable	Home-Based HIIE	MICE	*p*-Value
Total energy consumption (kcal)	99.5 ± 52.1	210 ± 43.3	<0.001
Mean %VO_2_max (%)	64.4 ± 7.4	49.3 ± 3.8	<0.001
Peak %VO_2_max (%)	81.1 ± 9.3	-	-
Mean %HRmax (%)	79.4 ± 4.9	66.2 ± 5.0	<0.001
Peak %HRmax (%)	88.9 ± 3.3	-	-
Rating of Perceived Exertion (RPE)	17 ± 1.8	11 ± 1.9	<0.001

Total energy consumption was calculated from the oxygen uptake and carbon dioxide production during each exercise. RPE is the median value of the measurements taken immediately after the end of the exercise.Abbreviations: HIIE, high-intensity interval exercise; MICE, moderate-intensity continuous exercise; VO_2_max, maximal oxygen uptake; HRmax, maximal heart rate.

**Table 3 ijerph-19-04227-t003:** Carbohydrate content.

Variable	H	M	C
Dinner on the day before each trial (g)	106.8 ± 27.1	95.9 ± 26.3	102.2 ± 25.7
Lunch on the day of each trial (g)	92.0 ± 37.1	86.8 ± 34.4	89.2 ± 35.9
Dinner on the day of each trial (g)	98.2 ± 20.3	96.9 ± 20.5	98.6 ± 20.5

There was no significant difference in any of the sugar content. Abbreviations: H, home-based high-intensity interval exercise (H condition); M, moderate-intensity continuous exercise (M condition); C, rest for 120 min (C condition).

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
