# Peer review of "Home-Based High-Intensity Interval Exercise Improves the Postprandial Glucose Response in Young Adults with Postprandial Hyperglycemia"

_ijerph, 2022, doi:10.3390/ijerph19074227_

Round 1

Reviewer 1 Report

Overall this was a well deigned study and well written paper. Please consider my comments below.

  1. Please comment on the percentage of carbohydrates in the experimental standardised meal compared to the percentage of carbohydrates in a typical meal.
  2. Please comment on the type of carbohydrate in the standardised meal (simple or complex).
  3. Please indicate the fiber percentage of the standardised meal.
  4. Please add BMI to Table 1.
  5. Please add a sentence indicating that although postprandially participants were hyperglycaemic, the average fasting glucose concentration was within normal range.
  6. Please provide an indication what is typically considered a small, medium and large effect size for partial eta-squared to assist readers interpretation of results.
  7. Please add that future studies should involve female participants.
  8. In the conclusion you state that "HBHIIE was able to correct postprandial hyperglycemia in approximately one-third of the time of MICE." This could be read as though the glucose concentration after HBHIIE reduced quicker than after MICE. Please reword for clarity.

Author Response

Dear Reviewer:

We appreciate the time and effort you have dedicated to providing insightful feedback on ways to strengthen our paper. We have incorporated changes that reflect the detailed suggestions you have graciously provided. We also hope that our edits and the responses we provide below satisfactorily address all the issues and concerns you and the reviewers have noted.

Reviewer 2 Report

The manuscript titled "High Intensity Interval Exercise at Home Improves Postmeal Glycemic Response in Young Adults with Postmeal Hyperglycemia" is very interesting.

It is known that controlled physical exercise can help diseases such as diabetes, ischemia by now, cardiovascular disorders.

The manuscript is well structured, however I have some clarifications to make to the authors:

1) Indicate before the title that it is an ARTICLE;

2) The legends of the figures are not very descriptive M, H, C goes to indicate what they are otherwise it becomes difficult to analyze.

3) Also, do the authors in the introduction talk about Covid-19? Because? Were these pre-covid patients not diabetic? Please explain better.

4) In addition to glycemia, fruit would also be interested in the levels of glycated hemoglobin,  and glycosuria of the patients ????? if they have not been evaluated explain why?

5) References regarding physical activity and how some biomarkers can be used for health monitoring

Author Response

(The authors gave the same response as above.)
